# Synthesis and Properties of Ethylene Imine-Based Porous Polymer Nanocomposites with Metal Oxide Nanoparticles

**DOI:** 10.3390/molecules30173574

**Published:** 2025-08-31

**Authors:** Naofumi Naga, Julia Janas, Tomoya Takenouchi, Tamaki Nakano

**Affiliations:** 1College of Engineering, Shibaura Institute of Technology, Toyosu Campus, 3-7-5 Toyosu, Koto-ku, Tokyo 135-8548, Japan; 2Graduate School of Science & Engineering, Shibaura Institute of Technology, Toyosu Campus, 3-7-5 Toyosu, Koto-ku, Tokyo 135-8548, Japan; 3MSc In Advanced Nano and Bio Materials MONABIPHOT, Wrocław University of Science and Technology, Wybrzeże Stanisława Wyspiańskiego 27, 50-370 Wrocław, Poland; 4Institute for Catalysis, Hokkaido University, N 21, W 10, Kita-ku, Sapporo 001-0021, Japan; tamaki.nakano@cat.hokudai.ac.jp; 5Integrated Research Consortium on Chemical Sciences, Institute for Catalysis, Hokkaido University, N 21, W 10, Kita-ku, Sapporo 001-0021, Japan

**Keywords:** poly(ethyleneimine), porous polymer, nanocomposite, silica, alumina, zirconia

## Abstract

Ethylene imine-based porous polymer nanocomposites were prepared by ring-opening polymerization of 2,2-bishydroxymethylbutanol-tris [3-(1-aziridinyl)propionate] (3AZ), a tri-functional aziridine compound, in the presence of commercially available metal oxide nanoparticles, SiO_2_ or ZrO_2_, accompanied by polymerization-induced phase separation. The reactions with SiO_2_ and ZrO_2_ nanoparticles successfully yielded nanocomposite porous polymers as rigid materials. The nanocomposite porous polymers with SiO_2_ and ZrO_2_ nanoparticles showed characteristic surface morphologies composed of gathered particles with diameters less than 1 micrometer. These nanocomposites were effective in increasing Young’s moduli of the porous polymers due to an increase in their bulk densities. The presence of SiO_2_ and ZrO_2_ nanoparticles in the porous polymers efficiently retarded thermal decomposition.

## 1. Introduction

The formation of a composite is a facile and effective method to improve various properties (including tensile strength, elastic modulus, heat distortion temperature, etc.) of a wide range of materials, such as ceramics and metal- and polymer-based materials. There are examples of polymer nanocomposites where polymer matrices and nanofillers are combined [1,2,3,4,5]. As polymer matrices, both thermoplastic resins, such as polyethylene, polypropylene, polyamide, and polyvinyl alcohol, and thermosetting resins, such as epoxy resin and phenol resin, are applicable. Representative nanofiller materials include inorganic nanoparticles, such as SiO_2_, ZrO_2_, and TiO_2_; layered materials, such as montmorillonite, graphene, and clay; and nanocarbons, such as carbon nanotubes, carbon black, and fullerene, with sizes of about 1 to 100 nm. These fillers can effectively improve the mechanical, thermal, gas barrier, electric, and optical properties of polymer matrices that alone show limited performance. In the preparation of nanocomposites, homogenous distribution of the fillers and controlled interfacial interaction between the polymer matrices and nanofillers are important to draw practical efficiency from fillers at small quantities.

On the other hand, porosification can remarkably modify the properties of polymer matrices. In this context, various types of porous polymers have been developed to attain specific features, such as lightness, specific surface area, and gas or liquid permeability [6,7,8]. Porous polymers can be used as monolithic columns, filters, membranes, supports, and insulators. Porosity may be introduced to polymers through template polymerization, post-polymerization, and phase separation (polymerization-induced phase separation, PIPS). PIPS is one of the most practical methods to prepare porous network polymers with a variety of molecular structures. Two types of phase separation processes are possible in PIPS: nucleation growth and spinodal decomposition processes. The spinodal decomposition process is widely detected in the formation process of various porous polymers, as shown in Figure 1. Phase separation is induced in the homogeneous reaction system, as shown in Figure 1a, by the formation of polymer networks. At the early stage of phase separation, a co-continuous structure is formed (Figure 1b). This structure transitions to particles by interfacial tension following the growth in their size, as shown in Figure 1c,d. When the phase separation of the reaction preferentially occurs, the reaction yields precipitation (Figure 1e). The morphology of the resulting porous polymer composite is determined by the fixed stage of the phase separation through network formation (polymerization).

Some porous polymers have been obtained by the ring-opening addition reaction of multi-functional epoxy and amine compounds with aliphatic and/or aromatic cyclic structures through PIPS [9,10,11,12,13,14,15,16,17]. Ring-opening reactions are considered more suited to the production of porous polymers because the reaction of the epoxy group forms an OH group, which should increase the polarity of the polymer chains being formed in the reaction and hence decrease their miscibility with non-polar organic solvents. Through such a mechanism, PIPS involving ring-opening reactions can successfully yield highly porous polymers.

In this work, we studied nanocomposite formation through PIPS. We believe that the method presented here can apply to a wide range of materials. As an example, we previously reported poly(methyl methacrylate) (PMMA) nanocomposite by a conventional radical polymerization of methyl methacrylate with polymerizable SiO_2_ nanoparticles in methanol. The resulting PMMA-SiO_2_ porous nanocomposites showed high mechanical strength, good thermal resistance, and coloration at the solvent-absorbed state due to the Christiansen effect [18]. Although this is a facile method to prepare porous polymer nanocomposite, a specific monomer containing polymerizable SiO_2_ nanoparticles is necessary. We found an extremely facile method to yield a porous polymer involving ring-opening polymerization of 3AZ that only requires dissolving 3AZ in water. Using this method, an ethyleneimine-based porous polymer was produced [19]. This reaction system is also applicable to preparing porous polymer composites with water-soluble cyclodextrins [20]. By contrast, composition with water-insoluble cyclodextrins is impossible due to the precipitation of the cyclodextrins during polymerization. Uniform dispersion and stability are necessary for nanocomposition using the polymerization system of 3AZ in water.

In this study, we aimed to develop a more widely applicable method using conventional and/or commercially available materials and examine the synthesis of ethyleneimine-based porous composite polymers by the ring-opening polymerization of 3AZ in commercially available metal oxide sols (aqueous colloidal metal oxides: SiO_2_ and ZrO_2_) (Figure 2). The effects of the metal oxides on the morphology, mechanical properties, and thermal stability of the resulting porous polymer composites were systematically studied.

## 2. Results and Discussion

### 2.1. Synthesis of 3AZ–Metal Oxide Porous Composite Polymers

Ring-opening polymerization of 3AZ was carried out in the presence of SiO_2_ nanoparticles in water. Features of the SiO_2_ sols are summarized in Figure 2, where ST-20: stabilized by sodium hydroxide, pH 10; ST-N: stabilized by ammonia, pH 10; ST-C: high-stability alkaline sol in neutral pH range, pH 9; ST-O: acidic sol (sodium ion reduced from Na type, no acid added), pH 3; and ST-AK: acidic sol (cationic surface), pH 5. Figure 1 shows the production diagrams of the reaction systems in the presence of different types of aqueous colloidal silica particles. The corresponding porous polymers were obtained under a low SiO_2_ nanoparticle feed. For reactions in the presence of basic SiO_2_ nanoparticles (ST-20, ST-N, or ST-C), high reaction temperatures were favorable to induce phase separation, yielding porous polymer composites with at most 6 wt% SiO_2_ nanoparticles. Low reaction temperatures and high SiO_2_ nanoparticle feed will decrease the phase separation rate, and the reaction system should be fixed before phase separation and yield gels. In the reactions with acidic SiO_2_ nanoparticles (ST-O or ST-AK), the corresponding porous polymer composites were obtained with a low SiO_2_ nanoparticle feed of 2 wt%. The presence of acidic SiO_2_ nanoparticles decreased the basicity of the reaction systems, which hindered the ring-opening polymerization of 3AZ. In the reactions with ST-AK, more than 4 wt% SiO_2_ nanoparticles tended to yield precipitates. One explanation for this result would be that the charge repulsion between the polymer networks and the cationic surface of the ST-AK SiO_2_ nanoparticles drastically decreased the miscibility between the composite and water.

### 2.2. Structure of 3AZ–Metal Oxide Porous Polymer Composites

The surface morphology of 3AZ-SiO_2_ porous polymer composites was observed by SEM. Figure 2 and Appendix A show SEM images of some representative porous polymer composites with SiO_2_ nanoparticles. Most of the porous polymer composites with SiO_2_ showed a surface morphology composed of connected particles, whose diameters were less than 1 μm, as summarized in Table 1. The 3AZ porous polymers without SiO_2_ nanoparticles showed larger particle sizes: 2.9 μm (at 20 °C) or 4.8 μm (at 40 °C). The porous structures of 3AZ-SiO_2_ polymer composites were formed by polymerization-induced phase separation via the spinodal decomposition process, as illustrated in Figure 1. The addition of SiO_2_ nanoparticles effectively transferred the co-continuous structure to the small particle structure, which was fixed before the growth of the particles. The porous colloid composite with 2.0 wt% ST-N showed a wrinkled surface, which was formed at the transition state from the co-continuous structure to the particle structure. These results indicate strong interaction between the polymer networks and SiO_2_ nanoparticles in the present reaction systems.

The particle size distribution was evaluated by standard deviation (SD) coefficient of validation (CV = SD/average diameter of the particles’ diameters) with histograms (Appendix A). All the porous polymer composites showed CV values ranging from 0.15 to 0.28, independent of the kind and concentration of SiO_2_ and reaction temperature. The results indicate that the presence of SiO_2_ does not affect the homogeneity of the reaction systems.

The specific surface area of the porous polymer was measured by nitrogen sorption. Although the adsorption isotherms of type V or III (IUPAC) were observed in some porous polymer composites (Appendix A), the surface area of some samples was too small (less than 5 m^2^/g) for quantitative evaluation. 

The molecular structure of a 3AZ-SiO_2_ (ST-20) porous polymer composite was studied by FT-IR spectroscopy, as shown in Figure 3. Both the absorption peaks derived from the 3AZ polymer; C=O stretching at 1730 cm^−1^, quaternary ammonium at 1590 cm^−1^, and SiO_2_ at 1100 cm^−1^ were detected in the spectrum of the porous polymer composite. These results demonstrate the formation of 3AZ polymer with the existence of SiO_2_ nanoparticles in the porous polymer composite. The interaction between the polymer networks and SiO_2_ nanoparticles induced a shift and/or shape change of the peaks. Although a clear peak shift cannot be observed in the comparison between Figure 3b (3AZ porous polymer) and Figure 3c (3AZ-SiO_2_ porous polymer composite), the shape change at 1200–1100 cm^−1^ may have been caused by sialylation of the amine moieties in the polymer networks.

The distribution of SiO_2_ colloid in a 3AZ-SiO_2_ porous polymer composite was studied by SEM-EDS. Figure 4 shows the SEM-EDX images of a porous polymer composite with SiO_2_ nanoparticles (ST-20, 4.0 wt%, run 9). The element mapping of Si showed a clear homogeneous distribution of SiO_2_ colloids in the porous polymer composite, as shown in Figure 4e. The composition ratio (atomic number ratio) of the C, N, O, and Si were 60.2%, 10.9%, 26.3%, and 2.6%, respectively. These ratios were close to the calculated theoretical values (63.6%, 9.0%, 24.2%, and 3.0% for C, N, O, and Si).

SEM images of 3AZ-ZrO_2_ porous polymer composites are summarized in Figure 5 and Appendix A. All the porous polymer composites showed a surface morphology composed of connected particles, whose diameters were less than 1 μm, as summarized in Table 2. The average particle size tended to decrease with increasing concentration of ZrO_2_ nanoparticles and polymerization temperature. These factors should increase the relative rate of polymerization (network formation) to the phase separation rate and fix the morphology at the early stage of phase separation, as shown in Figure 1c.

The particle size distribution was evaluated by SD and CV, as summarized in Table 2 (Appendix A). The CV values ranged from 0.13 to 0.23, which were lower than those of the 3AZ-SiO_2_ systems. Lower interaction between the polymer network and the ZrO_2_ nanoparticles would make it possible for the particle size distribution to be more homogeneous.

The specific surface area of some 3AZ-ZrO_2_ porous polymer composites was less than 5 m^2^/g, as observed in 3AZ-SiO_2_ porous polymer composites.

The molecular structure of 3AZ-ZrO_2_ porous polymer composites was also studied by FT-IR spectroscopy (Appendix A). In the FT-IR spectra of 3AZ-ZrO_2_ porous polymer composites, the absorption peaks derived from ZrO_2_ nanoparticles were detected at about 1160 cm^−1^ besides the peaks derived from the polymer networks formed by the ring-opening polymerization of 3AZ (Figure 3a). No shift changes or shape changes, which would be possible by the interaction between the polymer networks and ZrO_2_, were detected by the FT-IR spectroscopy.

SEM-EDX images of a porous polymer composite with ZrO_2_ nanoparticles (ZrO_2_, 4.0 wt%, run 38) also showed homogeneous distribution of the C, N, O, and Zr elements, and their composition ratios (atomic number ratio) were 54.5% (C), 10.6% (N), 32.9% (O), and 2.2% (Zr), respectively. These ratios roughly corresponded to the calculated theoretical values (63.6%, 9.0%, 24.2%, and 3.0% for C, N, O, and Si).

### 2.3. Mechanical and Thermal Properties of 3AZ–Metal Oxide Porous Polymer Composites

The mechanical properties of 3AZ-SiO_2_ porous polymer composites were evaluated by the compression test. Figure 6 shows the stress–strain curves of the porous polymer composites obtained with 2.0 wt% SiO_2_ colloids in the feed. The addition of SiO_2_ nanoparticles made the porous polymer significantly hard. The Young’s moduli of the porous polymer composites are summarized in Table 1, with the average particle size (diameter) and the bulk density. The 3AZ-SiO_2_ porous polymer composites were not breakable under the compression of 50 N. The primary particles of the composition with SiO_2_ nanoparticles decreased in size, which increased the bulk density. The SiO_2_ nanoparticles affected the mechanical properties of the 3AZ-SiO_2_ porous composite polymers, and the 3AZ-SiO_2_ porous polymer composite with ST-20 showed a higher Young’s modulus than the other 3AZ-SiO_2_ porous polymer composites. This may be due to the stabilizing of the SiO_2_ nanoparticles in the reaction systems. The electrical interaction between the SiO_2_ nanoparticle surface (Na^+ −^OSi-) and the amine moieties in the polymer networks of 3AZ (R_3_NH^+^) via ion exchange (R_3_NH^+ −^OSi- + Na^+^) preferentially occurred in the presence of Na^+^ in ST-20. The high Young’s moduli of the 3AZ-ST-C porous polymer composites would have been due to the high bulk densities. An increase in the SiO_2_ nanoparticle feed increased the Young’s moduli of the porous polymer composites. The combinations with SiO_2_ nanoparticles increased the rigidity of the porous polymer composites due to the increase in the bulk density and hardness of SiO_2_.

The mechanical properties of the 3AZ-ZrO_2_ porous polymer composites were also studied by compression test. Figure 7 shows the stress–strain curves of the porous polymer composites obtained with different ZrO_2_ nanoparticle feeds at 60 °C. The Young’s moduli of the porous polymer composites are summarized in Table 2, with the average particle size and bulk density. The Young’s moduli drastically increased in the polymer composites with more than 4.0 wt% ZrO_2_ nanoparticles. Increasing the ZrO_2_ nanoparticle concentration slightly decreased the Young’s moduli despite the increase in bulk density. In the same way, the porous polymer composites with 2.0 wt% or 4.0 wt% ZrO_2_ nanoparticles prepared under the polymerization temperature of 20 °C or 60 °C showed the highest Young’s modulus. One explanation for these results is that interfacial failure between polymer networks and ZrO_2_ nanoparticles would be dominant in porous polymer composites with higher ZrO_2_ nanoparticle concentrations. The porous polymer composites prepared at lower temperatures tended to show higher Young’s moduli. However, there was no clear correlation between the bulk density and Young’s modulus. One possible explanation for this is that porous polymer composites prepared at a lower temperature (20 °C) may have higher interaction between the ZrO_2_ nanoparticles and the polymer network, which would induce higher physical crosslinking density.

The distribution of the Young’s modulus was evaluated by Weibull analysis [21]. The Weibull moduli of the porous polymer composites, summarized in Table 1 and Table 2, ranged from 7 to 12, indicating moderate or large variation. The results were due to the characteristics of brittle materials.

The thermal stability of the 3AZ-SiO_2_ (ST-20, ST-O) porous polymer composites prepared at 60 °C was evaluated by TG-DTA analysis. The TG profiles of the 3AZ-ST-20 porous polymer composites are summarized in Figure 8. The 3AZ porous polymer (without SiO_2_) under an Ar atmosphere showed a two-step degradation at about 260 °C and 330 °C, caused by denaturation of amine groups and cleavage of ester bonds, respectively. The porous composite polymer with 2.0 wt% SiO_2_ nanoparticles showed almost the same profile as that of the 3AZ porous polymer. The porous polymer composites with 4.0 wt% and 6.0 wt% SiO_2_ nanoparticles mitigated the first decomposition and retarded the second decomposition. These phenomena were possibly due to effects such as inhibition of heat transfer and diffusion by SiO_2_ (barrier effect), restriction of the polymer chains’ motion (reduction of molecular mobility), formation of physical networks, and stabilization by carbonization residues (char formation). The thermal data from the TG profiles, onset temperature of thermal degradation (T_don_), maximum degradation rate (R_dmax_), and char yield at 500 °C of 3AZ-ST-20 porous polymer composites are summarized in Table 3. The increase in SiO_2_ concentration increased T_don_ and decreased R_dmax_. The char yield increased with increasing SiO_2_ concentration. The 3AZ-SiO_2_ porous polymer composite with 2.0 wt% acidic ST-O (run 24) showed similar T_don_ and R_dmax_ values as those of the corresponding 3AZ-ST-20 porous polymer composite (run 8). The results indicate that the features of SiO_2_ nanoparticles do not have much impact on the thermal degradation behavior.

Figure 9 shows the TG profiles of the 3AZ-ZrO_2_ porous polymer composites prepared at 60 °C. The porous polymer composites showed a multi-step degradation at more than 300 °C. These profiles are different from those of the porous composite polymer with SiO_2_ nanoparticles (Figure 8). These may be caused by differences in the interaction of the metal oxide used. In the case of nanocomposites with 3AZ polymer and SiO_2_ nanoparticles, the interaction would be mainly via hydrogen bonds derived from surface OH groups on the SiO_2_ nanoparticles. By contrast, the interaction via coordination and/or acid–base should be possible in nanocomposites with ZrO_2_ nanoparticles due to Lewis acidity. A difference in thermal conductivity between SiO_2_ (1.3 W/mK) and ZrO_2_ (2.5 W/mK) may affect the thermal decomposition behavior of the porous polymer composites [22,23]. The T_don_, R_dmax_, and char yield of the 3AZ-ZrO_2_ porous polymer composites with increasing ZrO_2_ concentration (Table 3) showed the same tendency as those with SiO_2_.

## 3. Materials and Methods

### 3.1. Materials

3AZ was kindly donated by Nippon Shokubai Co., Ltd., (Osaka, Japan) and used as received. Aqueous colloidal silicas (20 wt%, particle size 12 nm), SNOWTEX (ST-20: stabilized by sodium hydroxide, pH 10; ST-N: stabilized by ammonia, pH 10; ST-C: high-stability alkaline sol in neutral pH range, pH 9; ST-O: acidic sol (sodium ion reduced from Na type, no acid added), pH 3; and ST-AK: acidic sol (cationic surface), pH 5), were kindly donated by Nissan Chemical Corporation (Tokyo, Japan). A zirconia sol dispersed in water—NanoUse OZ-S20H: 20 wt%, particle size: 30–50 nm, weak acidity—was also kindly donated by Nissan Chemical Corporation. Methanol (MeOH) was commercially obtained from Kanto Chemical Co., Inc. (Tokyo, Japan) and used without further purification.

### 3.2. Synthesis of Porous Polymer Composites

The reaction of 20 wt% 3AZ in the presence of 2 wt% SiO_2_ nanoparticles is described as an example. Aqueous colloidal silica (0.40 g) and distilled water (2.8 g) were added to a 20 mL vial and stirred by a vortex mixer (Mixer N-40M-1, NISSIN, Tokyo, Japan) for a few minutes to make a diluted solution. Then, 3AZ (0.8 g) and the sol were added to a quadrangular prism polyethylene bottle (1.7 cm × 1.7 cm × 3.0 cm), stirred using a vortex mixer, and then stored at the desired temperature in an ESPEC SU-641 constant-temperature chamber (ESPEC CORP., Osaka, Japan) for 24 h. The obtained porous polymer composite was washed by immersion in excess of methanol for 24 h. The porous polymer composite was air-dried at room temperature for 24 h and further dried in vacuo at 40 °C for 3 h. Reactions with different metal oxides and concentrations were conducted by the same procedures.

### 3.3. Analytical Procedures

FT-IR spectra of the porous polymer composites were recorded on an FTIR-8400 or an IRAffinity-1S spectrometer (SHIMADZU Corporation, Kyoto, Japan), and 30 scans were accumulated from 4000 to 500 cm^−1^.

Scanning electron microscopy (SEM) images or SEM/energy-dispersive X-ray spectroscopy of 3AZ–metal oxide porous polymer composites were acquired by a JEOL JSM-7610F microscope (Tokyo, Japan) with an LEI detector at an acceleration voltage of 3.0 kV or 20 kV, respectively. The nitrogen (N) content was determined by the atomic number (Z), absorption (A), fluorescence (F) (ZAF) correction method. The average size in the SEM images was evaluated by image analysis using an Image-J software package (Version 1.54). About 50 to 100 particles were counted.

The surface area of the porous polymer was measured by nitrogen sorption using an Autosorb 6AG (Quantachrome Instruments, Boynton Beach, FL, USA) and a Belsorp II mini (MicrotracBEL Corp., Osaka, Japan). The samples were dried under reduced pressure at 100 °C for 1 h immediately before the measurement.

The mechanical properties of the porous polymer composites were investigated using the compression test with a Tensilon RTE-1210 apparatus (ORIENTEC Co., Ltd., Tokyo, Japan). The test samples were cut into a 1 cm^3^ cube and pressed at a rate of 0.5 mm/min at room temperature. The test was conducted five times per sample, and the middle result was adopted. The samples were stored at 23 °C and a relative humidity of 50% (JIS Z 8703) for 48 h before the test.

The bulk density of the porous polymer composite, g/cm^3^, was calculated from the weight of the samples before the test.

Thermogravimetric (TG) analysis of the porous polymer composites was conducted with a Bruker AXS TG-DTA2020SA (Billerica, MA, USA). The sample was heated from room temperature to 500 °C at a rate of 20 °C/min under an argon atmosphere.

## 4. Conclusions

The ring-opening polymerizations of 3AZ in water in the presence of SiO_2_ and ZrO_2_ nanoparticles successfully yielded ethylene imine-based polymer nanocomposites. The resulting porous polymer nanocomposites showed a morphology composed of connected particles with diameters less than 1 μm, which was formed through polymerization-induced phase separation via a spinodal decomposition process. The nanocomposites with SiO_2_ and ZrO_2_ nanoparticles had drastically decreased particle sizes and increased bulk densities and Young’s moduli. In the case of the 3AZ-SiO_2_ porous polymer nanocomposites, the nature of the SiO_2_ nanoparticles used affected the porous morphology and the mechanical properties. An increase in SiO_2_ nanoparticle concentration increased the Young’s moduli of the resulting porous polymer nanocomposites. ZrO_2_ nanoparticle concentrations of 2.0 or 4.0 wt% were suitable to attain 3AZ-ZrO_2_ porous polymer nanocomposites with high Young’s moduli. Further increase in ZrO_2_ nanoparticle concentration decreased the Young’s moduli of the porous polymer nanocomposites. One explanation for the result may be an interfacial failure between the polymer networks and the ZrO_2_ nanoparticles. The lower polymerization temperature increased the Young’s modulus of the porous polymer nanocomposites with ZrO_2_ nanoparticles. The nanocomposites with SiO_2_ and ZrO_2_ nanoparticles effectively retarded thermal decomposition.

The present research can provide a facile and practical method to prepare porous polymer nanocomposites. Here, a simple dissolution of 3AZ in commercially available metal oxide sols can yield the corresponding porous polymer nanocomposites. The previously reported 3AZ porous polymers (without metal oxide) were composed of micrometer-order particles, which showed high permeability of solutions. The features of 3AZ porous polymers should be suitable for separation and purification in biochemistry by chemical interactions. The present nanocomposition decreases the particle size in porous polymers to less than 1 μm. The size is less than that of suspended particulate matter. These porous polymer composites are applicable for separators and filters for soot, smoke, etc. by utilizing thermal stability. We are also preparing some porous polymer composites with other polymer materials using similar methods. These results will be reported elsewhere in due course.

## Data Availability

The data presented in this study are available in the article or in the Appendix A.

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
