# Peer review of "Synthesis and Properties of Ethylene Imine-Based Porous Polymer Nanocomposites with Metal Oxide Nanoparticles"

_molecules, 2025, doi:10.3390/molecules30173574_

Round 1

Reviewer 1 Report

Comments and Suggestions for Authors

This manuscript presents a straightforward method for synthesizing ethylene imine-based porous polymer nanocomposites using ring-opening polymerization of 3AZ in the presence of metal oxide nanoparticles (SiO2, Al2O3, ZrO2). The study demonstrates enhanced mechanical properties and thermal stability in SiO2 and ZrO2 composites, with potential applications in high-temperature filters. The work is innovative in its simplicity and use of commercially available materials. However, the characterization is notably superficial, lacking quantitative depth in morphological, structural, and mechanistic analyses. Key limitations include insufficient pore structure data, inadequate nanoparticle distribution validation, and speculative interpretations without experimental support. Addressing these gaps is essential to substantiate the claims and enhance the scientific impact.

  1. The manuscript omits critical quantitative data on pore size distribution, surface area, and porosity for the porous nanocomposites. BET surface area analysis, mercury intrusion porosimetry, or gas adsorption measurements must be performed to evaluate pore characteristics, as these directly influence mechanical and thermal properties. For example, the claimed "connected particles less than 1 μm" in SEM images (e.g., Figure 5) lacks statistical histograms or 3D tomography to confirm uniformity. Without this, correlations between morphology and properties (e.g., bulk density in Table 1) remain unsubstantiated.
  2. SEM analysis is restricted to selective samples, omitting failed systems like Al2O3 nanocomposites (mentioned only in supplementary Figure S1) and underrepresented conditions (e.g., low-temperature ZrO2 samples). High-resolution SEM images with scale bars should be provided for all nanocomposites, including Al2O3, to explain its fragility. Additionally, particle size data in Table 1 lack error ranges; image analysis with software like Image-J must include standard deviations from multiple regions to ensure reliability.
  3. FT-IR spectra (e.g., Figure 3 and S2) only confirm functional group presence but fail to quantify interactions like hydrogen bonding or Lewis acid-base coordination between nanoparticles and polymer. XPS should be employed to analyze surface chemistry (e.g., Si 2p or Zr 3d peaks) and covalent bonding. FT-IR mapping could also visualize nanoparticle dispersion homogeneity, as current EDX data (Figure 4) are limited to one sample and lack elemental quantification.
  4. Young's modulus data (Tables 1 and 2) report no standard deviations or replicate analyses, undermining statistical significance. Error bars must be added to stress-strain curves (e.g., Figure 6 and 7), and compression tests should include ≥3 replicates per condition. The hypothesis of "interfacial failure" causing modulus reduction in high-ZrO2 composites (Section 3.3) requires TEM or AFM to directly observe nanoparticle-polymer interfaces and crack propagation.
  5. TG profiles (Figures 8 and 9) are interpreted via "barrier effects" or "molecular mobility restriction," but no mass spectrometry (MS) data identify decomposition gases or residues. TGA-MS should be performed to validate degradation steps (e.g., amine denaturation at 260°C). Additionally, char yield analysis is missing; quantifying residue mass could support claims of thermal retardation by nanoparticles.
  6. Elemental mapping via SEM-EDX is only exemplified in Figure 4 for SiO2, neglecting ZrO2 and Al2O3 systems. Quantitative EDX or TEM should be applied to all nanocomposites to measure nanoparticle dispersion homogeneity and local composition. Zeta potential or DLS of initial sols could also correlate colloidal stability (e.g., pH effects) with final morphology.
  7. The influence of nanoparticle stabilizers (e.g., HNO3 in Al2O3) on polymerization is discussed qualitatively, but no pH measurements of reaction mixtures are provided. Real-time pH monitoring or zeta potential analysis should be included to mechanistically explain phase separation failures (e.g., Al2O3 fragility) and success with SiO2/ZrO2.
  8. Potential uses as high-temperature filters are mentioned, yet no functional data (e.g., filtration efficiency, pressure drop, or contaminant adsorption) are presented. Bench-scale filtration tests under controlled conditions must be conducted to validate practical relevance and justify claims in the conclusion.

Author Response

For Review 1:

1&2. The specific surface area and pore size distribution of the porous polymers was evaluated by the Brunauer-Emmett-Teller (BET) and BJH methods. However, the surface area of some samples was too small, less than 5 m2/g, for quantitative evaluation. The low specific surface area may have been derived not from a “microporous,” but rather from a “macroporous” structure, as previously reported for other porous polymers.

The particle size distribution was evaluated by standard deviation (SD) coefficient of validation (CV = SV/average diameter of the particles’ diameters) with histograms (Figure S2).

Mercury porosimeter and gas perimetry are under consideration in the next research for the applications.

All the data of 3AZ-Al2O3 system are deleted in the revised version. The SEM images in the text are carefully selected and scale bars are added also to those of 3AZ-ZrO2 porous polymer composites.

  1. The interaction between the polymer networks and SiO2 nanoparticles should induce shift and/or shape change of the peaks. Although clear peak’s shift was not observed in the comparison between Figure (b) AZ porous polymer and Figure (c) 3AZ-SiO2, the shape change at 1200-1100 cm-1 may be derived from sialylation of the amine moieties in the polymer networks. The shift changes nor the shape change, which would be possible by the interaction between the polymer networks and ZrO2, was not detected the FT-IR spectroscopy.

We understand the usefulness of XPS and FT-IR mapping. But unfortunately, we don’t have them. We would like to try (or outsource) these analyses in the next research for the applications.

EDX analyses were applied to both the representative samples, and the data was added to the revised manuscript.

  1. The test was conducted five times per sample, and the middle result was adopted.

TEM should be useful for direct observation. However, we could not prepare the preparation of the specimen, slicing of the porous polymer composites. We don’t have AFM. We would like to try (or outsource) the analysis in the next research for the applications.

The ‘interfacial failure’ is an “one explanation” for the result. The sentence was revised to avoid misunderstanding.

  1. We understand the usefulness of TGA-MS. But unfortunately, we don’t have it. We would like to try (or outsource) the analysis in the next research for the applications.

  1. EDX analyses were applied to the representative samples, and the data was added to the revised manuscript.

Due to the contract with the company that provided the samples, detailed analysis of the samples is prohibited.

  1. We tried to real-time pH monitor of the reaction system. However, the phase separation in the reaction system hindered the contact of the pH sensor and did not show accurate values.

  1. The previously reported 3AZ porous polymers (without metal oxide) were composed of micrometer-order particles, which showed high permeability of solutions. The features of the 3AZ porous polymers should be suitable for separation and/purification in biochemistry by chemical interactions. The present nano composition decreases the particle size in the porous polymers, less than 1 mm. The size is less than that of suspended particulate matter. These porous polymer composites are applicable for separators and filters for soot and smoke etc., by utilizing thermal stability.

These sentences were added in Conclusion of the revised manuscript.

Reviewer 2 Report

Comments and Suggestions for Authors

1. Clarify the novelty of the work compared to your previous studies on 3AZ-based porous polymers (Refs. 19–21) and PMMA–SiO₂ nanocomposites (Ref. 22). The introduction mentions these works but does not clearly state how the current study advances beyond them.

2. Provide a stronger justification for selecting SiO₂, Al₂O₃, and ZrO₂ nanoparticles—are they chosen for specific interactions, availability, or anticipated performance differences?

3. The introduction contains long lists of examples (lines 34–41, 38–40) which interrupt the logical flow. Consider condensing them and focusing on the gap in knowledge you are addressing.

4. The mechanism of polymerization-induced phase separation (PIPS) is described but scattered; move the mechanistic explanation to a coherent paragraph, possibly illustrated with a schematic earlier in the introduction.

5. Justify the choice of reaction conditions (3AZ concentration, nanoparticle loading ranges, temperatures). Are these based on optimization or previous work?

6. For the various SiO₂ sols (ST-20, ST-N, ST-C, ST-O, ST-AK), briefly explain the significance of their surface chemistry and pH differences before discussing results, so the reader understands why their behavior differs.

7. Include details on how the “average particle size” from SEM was determined—number of particles counted, measurement method, and statistical uncertainty.

8. The discussion of acidic vs. basic SiO₂ sols (lines 142–148) should be supported by quantitative data on polymerization rates or phase separation kinetics, not only qualitative observations.

9. The proposed spinodal decomposition mechanism (Scheme 2) is plausible, but no direct evidence (e.g., time-resolved morphology) is shown. Consider citing literature or adding experimental support for this mechanism.

10. Discuss more deeply why ST-20 and ST-C gave particularly high moduli compared to other SiO₂ sols—link to their stabilizers or surface chemistry.

11. For ZrO₂ composites, the explanation involving “interfacial failure” at higher loadings is speculative; consider adding fracture surface SEM images or adhesion-related testing to support this claim.

12. TG curves (Figs. 8 and 9) are interpreted mainly in qualitative terms. Consider quantifying onset temperatures (T₅%, T₁₀%), maximum degradation rates, and char yields to strengthen the conclusions.

13. The discussion on differences between SiO₂ and ZrO₂ interactions (lines 306–311) is interesting but needs literature references to support claims about hydrogen bonding vs. Lewis acidity effects.

14. The conclusions section largely repeats results without broader implications. Discuss how these findings could influence future applications (filters under high temperatures, as briefly mentioned) and potential scalability or industrial relevance.

15. Acknowledge limitations—e.g., fragility of Al₂O₃ composites—and suggest how these could be addressed in future work.

16. Remove minor redundancies, e.g., repeated FT-IR descriptions in the text and figure captions.

17. Standardize units (µm vs. micrometer, °C formatting, wt% formatting).

Author Response

For Review 2:

1&3. The introduction was revised accordingly.

  1. The reason for selecting these metal oxides is commercial availability.

  1. The mechanism of PIPS was explained in the introduction for the benefit of the readers accordingly

  1. The reaction conditions of present study were based on the previous report without metal oxide (ref 18).

  1. According to the reviewer’s suggestion, the explanation of SiO2 sols was added in the beginning of the “Results and Discussion”.

  1. The average size in the SEM images was evaluated by image analysis using an Image-J software package. About 50 to 100 particles were counted. The particle size distribution was evaluated by standard deviation (SD) coefficient of validation (CV = SV/average diameter of the particles’ diameters) with histograms.

These sentences are added in the revised manuscript.

  1. We tried to evaluate the phase separation rate to monitor the transmitted light intensity of the reaction system as previously reported in ref 19. However, the transmitted light intensity decreased in the presence of metal oxides, which was insufficient for quantitative evaluation.

  1. We previously tried to trace the phase separation process by termination of the reaction during the phase separation in a 3AZ polymerization. The phase separation rate was to fast to trace the transition state of the reaction by the method.

  1. The electrical interaction between the SiO2 nanoparticles surface (Na+ -OSi-) and the amine moieties in the polymer networks of 3AZ (R3NH+) via ion exchange (R3NH+ -OSi- + Na+) should preferentially occurs in the presence of Na+ in ST-20. The high Young’s moduli of the 3AZ-ST-C pours polymer composites would be derived from the high bulk densities.

These sentences are added in the revised manuscript.

  1. The ‘interfacial failure’ is an “one explanation” for the result and has not confirmed yet. The sentence was revised to avoid misunderstanding.

  1. The qualitative data are acquired from the TG curves and summarized in Table 3 to evaluate the effect of the metal oxide concentration qualitatively.

  1. The references were added in the revised manuscript.

  1. The previously reported 3AZ porous polymers (without metal oxide) were composed of micrometer-order particles, which showed high permeability of solutions. The features of the 3AZ porous polymers should be suitable for separation and/purification in biochemistry by chemical interactions. The present nano composition decreases the particle size in the porous polymers, less than 1 mm. The size is less than that of suspended particulate matter. These porous polymer composites are applicable for separators and filters for soot and smoke etc., by utilizing thermal stability.

These sentences were added in Conclusion of the revised manuscript.

  1. All the data of 3AZ-Al2O3 system are deleted in the revised version. More detailed studies must be necessary to report the results.

16&17. Revised accordingly.

Round 2

Reviewer 1 Report

Comments and Suggestions for Authors

The authors have addressed key revisions effectively, but some issues require refinement to meet Molecules' standards for mechanistic clarity, reproducibility, structural characterization, and editorial precision:

  1. Compression tests lack reproducibility safeguards. Modulus variations (e.g., Run 29 vs. 30) omit strain rates, hydration states, and preconditioning. Standardize 48-hr humidity equilibration (30% RH) and report Weibull distribution analysis (≥5 replicates) to ensure reliability for filter/separator applications.
  2. Pore structure analysis requires rigorous N₂ physisorption characterization. The dismissal of BET/BJH results due to "low surface area" (<5 m²/g, Section 2.2) overlooks critical insights from adsorption-desorption isotherms and pore size distribution plots. These are essential to: Distinguish meso/macropore dominance (e.g., hysteresis loop shape indicating pore connectivity); Correlate pore geometry with permeability claims (Lines 385–390); Benchmark against literature (e.g., ACS Catal. 2024, 14, 10245; Chem. Eng. J. 2024, 492, 152322). Include isotherms and BJH plots for all composites to validate morphological differences.
  3. Figure 3’s caption contains a critical editorial error: "Figure 3. FT-IR spectra of FT-IR spectra of (a) 3AZ monomer..."
  4. The manuscript's attribution of mechanical enhancement solely to increased bulk density (Tables 1–2) is contradicted by inverse trends. e.g., higher-density composites (ST-C at 4 wt%: 0.812 g/cm³) exhibit lower Young’s moduli (3,952 kPa) than lower-density counterparts (ST-20: 0.660 g/cm³, 7,611 kPa), necessitating reanalysis via nanoparticle dispersion efficacy (e.g., SEM particle connectivity in Figure 2/5) and interface integrity to resolve discrepancies.
  5. Claims of nanoparticle-driven thermal stability lack mechanistic specificity, as TGA data (Figures 8–9) reveal unexplained divergences. e.g., acidic SiO₂ (ST-O, pH 3) shows higher degradation rates (Rdmax = 1.07 wt%/K) than basic SiO₂ (ST-20, pH 10) at identical loading, demanding explicit links between nanoparticle surface chemistry (pH, charge) and degradation kinetics using existing characterization (FT-IR, SEM-EDX).

Author Response

  1. The samples were stored under the conditions at 23 °C and relative humidity of 50% (JIS Z 8703) for 48 h before the test.

The test samples were cut into a 1 cm3 cube and pressed at a rate of 0.5 mm/min at room temperature.

Distribution of the Young’s modulus was evaluated by Weibull analysis. The Weibull moduli of the porous polymer composites, summarized in Tables 1 and 2, ranged from 7 to 12 indicating moderate or large variation. The results should be derived from the characteristics of brittle materials.

The references were added to the revised manuscript.

  1. The specific surface area of the porous polymer composite was measured by nitrogen sorption. Although the adsorption isotherms of type V or III (IUPAC) were observed in some porous polymer composites (Figure S6), the surface area of some samples was too small, less than 5 m2/g, for quantitative evaluation.

 These results would be caused by weak interactions between nitrogen and the surface of the porous polymers. We would like to try to analyze it by another suitable method in the next research for the applications.

  1. Thank you for pointing it out. The caption was collected accordingly.

  1. Thank you for pointing this out. We miss-typed the Young’s modulus of run 18.

  1. A 3AZ- SiO2 porous polymer composite with 2.0 wt% of acidic ST-O (run 24) showed similar Tdon and Rdmax values to those of the corresponding 3AZ- ST-20 porous polymer composite (run 8). The results indicate that the features of the SiO2 nanoparticles do not have much impact on the thermal degradation behavior.

The references were added to the revised manuscript.

Reviewer 2 Report

Comments and Suggestions for Authors

Accept

Author Response

Thank you for the acceptance of our submission. Your comments were valuable to revise the manuscript.